# Detecting Extrapolation with Local Ensembles

**David Madras**[*]
University of Toronto
Vector Institute
madras@cs.toronto.edu

**James Atwood**
Google Brain
atwoodj@google.com

**Alex D'Amour**
Google Brain
alexdamour@google.com

## Abstract

We present *local ensembles*, a method for detecting extrapolation at test time in a pre-trained model. We focus on *underdetermination* as a key component of extrapolation: we aim to detect when many possible predictions are consistent with the training data and model class. Our method uses local second-order information to approximate the variance of predictions across an ensemble of models from the same class. We compute this approximation by estimating the norm of the component of a test point's gradient that aligns with the low-curvature directions of the Hessian, and provide a tractable method for estimating this quantity. Experimentally, we show that our method is capable of detecting when a pre-trained model is extrapolating on test data, with applications to out-of-distribution detection, detecting spurious correlates, and active learning.

## 1 Introduction

As machine learning is deployed in increasingly vital areas, there is increasing demand for metrics that draw attention to potentially unreliable predictions. One important source of unreliability is *extrapolation*. Extrapolation can be formalized in a number of ways: it can refer to making predictions on inputs outside the support of the training data, making predictions with high Bayesian or Frequentist uncertainty, or making predictions that depend strongly on arbitrary choices outside of the learning problem specification (e.g., a random seed). In this paper, we develop a method for detecting this last form of extrapolation. Specifically, we say that a trained model is extrapolating on a test input if the prediction at this input is *underdetermined* — meaning that many different predictions are all equally consistent with the constraints posed by the training data and the learning problem specification (i.e., the model architecture and the loss function).

Underdetermination is just one form of extrapolation, but it is particularly relevant in the context of overparameterized model classes (e.g. deep neural networks). Recently, simple (but computationally expensive) ensembling methods (Lakshminarayanan et al., 2017), which train many models on the same data from different random seeds, have proven highly effective at uncertainty quantification tasks (Ovadia et al., 2019). This suggests that underdetermination is a key threat to reliability in deep learning, and motivates flexible methods that can detect underdetermined predictions cheaply.

With this motivation, we present *local ensembles*, a post-hoc method for measuring the extent to which a pre-trained model's prediction is underdetermined for a particular test input. Given a trained model, our method returns an extrapolation score that measures the variability of test predictions across a *local ensemble*, i.e. a set of local perturbations of the trained model parameters that fit the training data equally well. Local ensembles are a computationally cheap, post-hoc alternative to fully trained ensembles, and do not require special training procedures of approximate ensembling methods that measure related, but distinct, notions of uncertainty (Gal & Ghahramani, 2015; Blundell et al., 2015). Local ensembles also address a gap in approximate methods for estimating prediction uncertainty. Specifically, whereas exact Bayesian or Frequentist uncertainty includes underdetermination as one component, approximate methods such as Laplace approximations (MacKay, 1992) or influence function-based methods (Schulam & Saria, 2019) break down when underdetermination is present.

---

[*]Some of this work was done while this author was an intern at Google Brain.

In contrast, our method leverages the pathology that makes these methods struggle (an ill-conditioned Hessian).

Our contributions in this paper are as follows:

- We present *local ensembles*, a test-time method for detecting underdetermination-based extrapolation in overparameterized models.
- We demonstrate theoretically that our method approximates the variance of a trained ensemble with local second-order information.
- We give a practical method for tractably approximating this quantity, which is simpler and cheaper than alternative second-order reliability methods.
- Through experiments aimed at testing underdetermination, we show our method approximates the behavior of trained ensembles, and can detect extrapolation in a range of scenarios.

## 2 EXTRAPOLATION SCORE AND LOCAL ENSEMBLES

### 2.1 SETUP

Let $z = (x, y)$ be an example input-output pair, where $x$ is a vector of features and $y$ is a label. We define a model in terms of a loss function $\mathcal{L}$ with parameters $\theta$ as a sum over training examples $(z_i)_{i=1}^n$, i.e., $\mathcal{L}(\theta) = \sum_i^n \ell(z_i, \theta)$, where $\ell$ is an example-wise loss (e.g., mean-squared error or cross entropy). Let $\theta^\star$ be the parameters of the trained model, obtained by, e.g., minimizing the loss over this dataset, i.e., $\theta^\star = \arg\min_\theta \mathcal{L}(\theta)$. We write the prediction function given by parameters $\theta$ at an input $x$ as $\hat{y}(x, \theta)$. We consider the problem of auditing a trained model, where unlabeled test points $x'$ arrive one at a time in a stream, and we wish to assess extrapolation on a point-by-point basis.

In this section, we introduce our local ensemble extrapolation score $\mathcal{E}_m(x')$ for an unlabeled test point $x'$ (significance of $m$ explained below). The score is designed to measure the variability that would be induced by randomly choosing predictions from an ensemble of models with similar training loss. Our score has a simple form: it is the norm of the prediction gradient $g_{\theta^\star}(x') := \nabla_\theta \hat{y}(x', \theta^\star)$ multiplied by a matrix of Hessian eigenvectors spanning a subspace of low curvature $U_m$ (defined in more detail below).

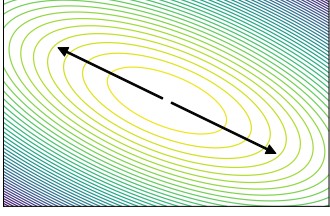

$$\mathcal{E}_m(x') = \|U_m^\top g_{\theta^\star}(x')\|_2 \tag{1}$$

Here, we show that this score is proportional to the standard deviation of predictions across a local ensemble of models with near-identical training loss, and demonstrate that this approximation holds in practice.

Figure 1: In this quadratic bowl, arrows denote the small eigendirection, where predictions are slow to change. We argue this direction is key to extrapolation.

### 2.2 DERIVATION

Our derivation proceeds in two steps. First, we define a local ensemble of models with similar training loss, then we state the relationship between our extrapolation score and the variability of predictions within this ensemble.

The spectral decomposition of the Hessian $H_{\theta^\star}$ plays a key role in our derivation. Let

$$H_{\theta^\star} = U\Lambda U^\top, \tag{2}$$

where $U$ is a square matrix whose columns are the orthonormal eigenvectors of $H_{\theta^\star}$, written $(\xi_{(1)}, \cdots, \xi_{(p)})$, and $\Lambda$ is a square, diagonal matrix with the eigenvalues of $H_{\theta^\star}$, written $(\lambda_{(1)}, \cdots, \lambda_{(p)})$, along its diagonal. As a convention, we index the eigenvectors and eigenvalues in decreasing order of the eigenvalue magnitude.

To construct a local ensemble of loss-preserving models, we exploit the fact that eigenvectors with large corresponding eigenvalues represent directions of high curvature, whereas eigenvectors with small corresponding eigenvalues represent directions of low curvature. In particular, under the

assumption that the model has been trained to a local minimum or saddle point, parameter perturbations in flat directions (those corresponding to small eigenvalues $\lambda_{(j)}$) do not change the training loss substantially (see Fig. 1). [1] We characterize this subspace by the span of eigenvectors with corresponding small eigenvalues. Formally, let $m$ be the eigenvalue index such that the eigenvalues $\{\lambda_{(j)} : j > m\}$, are sufficiently small to be considered "flat". [2] We call the subspace spanned by $\{\xi_{(j)} : j \in \sigma\}$ the *ensemble subspace*. Parameter perturbations in the ensemble subspace generate an ensemble of models with near-identical training loss. [3]

Our score $\mathcal{E}_m(x')$ characterizes the variance of test predictions with respect to random parameter perturbations $\Delta_\theta$ that occur in the ensemble subspace. We now show that our extrapolation score $\mathcal{E}_m(x')$ is, to the first order, proportional to the standard deviation of predictions at a point $x'$.

**Proposition 1.** *Let $\Delta_\theta$ be a random parameter perturbation with mean zero and covariance proportional to the identity $\epsilon \cdot I$, projected into the ensemble subspace spanned by $\{\xi_{(j)} : j > m\}$. Let $P_\Delta(x')$ be the linearized change in a test prediction at $x'$ induced by perturbing the learned parameters $\theta^\star$ by $\Delta_\theta$:*

$$P_\Delta(x') := g_{\theta^\star}(x')^\top \Delta_\theta \approx \hat{y}(x', \theta^\star + \Delta_\theta) - \hat{y}(x', \theta^\star).$$

*Then $\mathcal{E}_m(x') = \epsilon^{-1/2} \cdot SD(P_\Delta(x'))$.*

*Proof.* First, we characterize the distribution of the loss-preserving parameter perturbation $\Delta_\theta$. Let $U_m$ be the matrix whose columns $\{\xi_{(j)} : j > m\}$ span the ensemble subspace. Then $U_m U_m^\top$ is a projection matrix that projects vectors into the ensemble subspace, so the projected perturbation $\Delta_\theta$ has covariance $\epsilon \cdot U_m U_m^\top$. From this, we derive the variance of the linearized change in prediction

$$Var(P_\Delta) = \epsilon \cdot g_{\theta^\star}(x')^\top U_m U_m^\top g_{\theta^\star}(x') = \epsilon \cdot \|U_m^\top g_{\theta^\star}(x')\|_2^2 = \epsilon \cdot \mathcal{E}_m(x')^2. \qquad \square$$

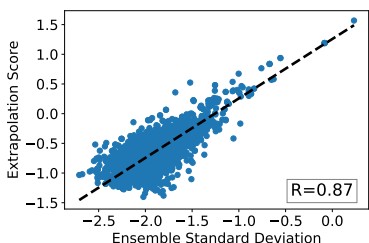

Figure 2: Mean extrapolation score vs. ensemble prediction standard deviation on WineQuality dataset. Shows line of best fit and Pearson coefficient R. Both axes log-scaled.

We test this hypothesized relationship on several tabular datasets. We train an ensemble of twenty neural networks using the same architecture and training set, varying only the random seed. Then, for each model in the ensemble, we calculate our local ensemble extrapolation score $\mathcal{E}_m(x)$ for each input $x'$ in the test set (see Sec. 3 for details). For each $x'$, we compare, across the ensemble, the mean value of $\mathcal{E}_m(x')$ to the standard deviation $\hat{y}(x')$. In Fig. 2, we plot these two quantities against each other for one of the datasets, finding a nearly linear relationship. On each dataset, we found a similar, significantly linear relationship (see Table 1 and Appendix A). We note the relationship is weaker with the Diabetes dataset; the standard deviations of these ensembles are an order of magnitude higher than the other datasets, indicating much noisier data. Finally, we note that we can obtain similar results for the standard deviation of the loss at test points if we redefine $g$ as the loss gradient rather than the prediction gradient.

## 3 COMPUTING EXTRAPOLATION SCORES

---

[1] To see this, consider a second-order Taylor approximation to the training loss when the parameters are perturbed by a vector $\Delta_\theta$: $\mathcal{L}(\theta^\star + \Delta_\theta) - \mathcal{L}(\theta^\star) \approx \frac{1}{2}\Delta_\theta^\top H_{\theta^\star}\Delta_\theta = \sum_{j=1}^{p} \lambda_{(j)} \left(\xi_{(j)}^\top \Delta_\theta\right)^2$. Note that the first-order term drops out because we assume $\theta^\star$ lies at a local minimum or saddle point. Perturbations $\Delta_\theta$ that lie in the subspace spanned by eigenvectors $\xi_{(j)}$ whose corresponding eigenvalues $\lambda_{(j)}$ are close to zero contribute negligibly to this sum.

[2] For now, we take $m$ to be given, and discuss tradeoffs for choosing $m$ in practice in Section 3.

[3] We note that not all loss-preserving ensembles will necessarily be local; however, the existence of some such local ensemble is sufficient to enable underdetermination.

We now discuss a practical method for computing our extrapolation scores. The key operation is constructing the set of eigenvectors that span a suitably loss-preserving ensemble subspace (i.e. have sufficiently small corresponding eigenvalues). Because eigenvectors corresponding to large eigenvalues are easier to estimate, we construct this subspace by finding the top $m$ eigenvectors and defining the ensemble subspace as their orthogonal complement. Our method can be implemented with any algorithm that returns the top $m$ eigenvectors. See below for discussion on some tradeoffs in the choice of $m$, as well as the algorithm we choose (the Lanczos iteration (Lanczos, 1950)).

| Dataset | Pearson |
|---------|---------|
| Boston | 0.76 |
| Diabetes | 0.50 |
| Abalone | 0.76 |
| Wine | 0.87 |

Table 1: Correlation of extrapolation scores and ensemble std. deviations on 4 datasets.

Our method proceeds as follows. For a given choice of $m$, we calculate the $m$ eigenvectors of $H$ with the *largest* eigenvalues. These eigenvectors define an $m$-dimensional subspace which is the orthogonal complement to the ensemble subspace. We use these eigenvectors to construct a matrix that projects gradients into the ensemble subspace. Specifically, let $U_{m\perp}$ be the matrix whose columns are these large-eigenvalue eigenvectors $\{\xi_{(j)} : j \leq m\}$. Then $U_{m\perp} U_{m\perp}^{\top}$ is the projection matrix that projects vectors into the "large eigenvalue" subspace, and $I - U_{m\perp} U_{m\perp}^{\top}$ projects into its complement: the ensemble subspace. Now, for any test input $x'$, we take the norm of this projected gradient to compute our score

$$\mathcal{E}_m(x') = \left\| \left( I - U_{m\perp} U_{m\perp}^{\top} \right) g_{\theta^\star}(x') \right\|_2 .$$

The success of this approach depends on the choice of $m$. Specifically, the extrapolation score $\mathcal{E}_m(x')$ is the most sensitive to underdetermination in the region of the trained parameters if we set $m$ to be the smallest index for which the training loss is relatively flat in the implied ensemble subspace. If $m$ is set too low, the ensemble subspace will include well-constrained directions, and $\mathcal{E}_m(x')$ will over-estimate the prediction's sensitivity to loss-preserving perturbations. If $m$ is set too high, the ensemble subspace will omit some under-constrained directions, and $\mathcal{E}_m(x')$ will be less sensitive. For models where all parameters are well-constrained by the training data, a suitable $m$ may not exist. This will usually not be the case for deep neural network models, which are known to have very ill-conditioned Hessians (see, e.g., Sagun et al., 2017). Whether a direction is "well-constrained" is ultimately a judgment call for the user. One potential heuristic is to consider a small parameter perturbation of a fixed norm in the ensemble subspace, and to set a tolerance for how much that perturbation can change the training loss; for small perturbations, the change in loss is a linear function of the curvature in the direction of the perturbation, which is upper bounded by $\lambda_{(m)}$.

We use the Lanczos iteration to estimate the top $m$ eigenvectors, which presents a number of practical advantages for usage in our scenario. Firstly, it performs well under early stopping, returning good estimates of the top $m$ eigenvectors after $m$ iterations. Secondly, we can cache intermediate steps, meaning that computing the $m + 1$-th eigenvector is fast once we have computed the first $m$. Thirdly, it requires only implicit access to the Hessian through a function which applies matrix multiplication, meaning we can take advantage of efficient Hessian-vector product methods (Pearlmutter, 1994).

Finally, the Lanczos iteration is simple – it can be implemented in less than 20 lines of Python code (see Appendix B.1). It contains only one hyperparameter, the stopping value $m$. Fortunately, tuning this parameter is efficient — given a maximum value $M$, we can try many values $m < M$ at once, by estimating $M$ eigenvectors and then calculating $\mathcal{E}_m$ by using the first $m$ eigenvectors. The main constraint of our method is space rather than time — while estimating the first $m$ eigenvectors enables easy caching for later use, it may be difficult to work with these eigenvectors in memory as $m$ and model size $p$ increase. This tradeoff informed our choice of $m$ in this paper; we note in some cases that increasing $m$ further could have improved performance (see Appendix E). This suggests that further work on techniques for mitigating this tradeoff, e.g. online learning of sparse representations (Wang & Lu, 2016; Wang et al., 2012), could improve the performance of our method. See Appendix B for more details on the Lanczos iteration.

## 4 RELATED WORK

### 4.1 RELATION TO BAYESIAN AND FREQUENTIST SECOND-ORDER METHODS

It is instructive to compare our extrapolation score to two other approximate reliability quantification methods that are aimed at Bayesian and Frequentist notions of extrapolation, respectively. Like our extrapolation score, both of these methods make use of local information in the Hessian to make an inference about the variance of a prediction. First, consider the Laplace approximation of the posterior predictive variance. This metric is derived by interpreting the loss function as being equivalent to a Bayesian log-posterior distribution over the model parameters $\theta$, and approximating it with a Gaussian. Specifically, (see, e.g., MacKay, 1992)

$$Var(y \mid x') \approx g_{\theta^\star}(x')^\top H_{\theta^\star}^{-1} g_{\theta^\star}(x') = \sum_{j=1}^{p} \lambda_{(j)}^{-1} \left( \xi_{(j)}^\top g_{\theta^\star}(x') \right)^2. \tag{3}$$

Second, consider scores such as RUE (Resampling Under Uncertainty) designed to approximate the variability of predictions by resampling the training data (Schulam & Saria, 2019). These methods approximate the change in trained parameter values induced by perturbing the training data via influence functions (Koh & Liang, 2017; Cook & Weisberg, 1982). Specifically, the gradient of the parameters with respect to the weight of a given training example $z_i$ is given by

$$I(z_i) = -H_{\theta^\star}^{-1} \nabla_\theta \ell(z_i, \theta^\star) = \sum_{j=1}^{p} \lambda_{(j)}^{-2} \left( \xi_{(j)}^\top \nabla_\theta \ell(z_i, \theta^\star) \right)^2. \tag{4}$$

Schulam & Saria (2019) combine this influence function with a specific random distribution of weights to approximate the variance of predictions under bootstrap resampling; other similar formulations are possible, sometimes with theoretical guarantees (Giordano et al., 2019).

Importantly, both of these methods work well when model parameters are well-constrained by the training data, but they struggle when predictions are (close to) underdetermined. This is because, in the presence of underdetermination, the Hessian becomes ill-conditioned. Practical advice for dealing with this ill-conditioning is available (Koh & Liang, 2017), but we note that this not merely a numerical pathology; by our argument above, a poorly conditioned Hessian is a clear signal of extrapolation. In contrast to these methods, our method focuses specifically on prediction variability induced by underconstrained parameters. Our extrapolation score incorporates *only* those terms with small eigenvalues, and removes the inverse eigenvalue weights that make inverse-Hessian methods break down. This is clear from its summation representation: $\mathcal{E}_m(x') = \sum_{j>m} \left( \xi_{(j)}^\top g_{\theta^\star}(x') \right)^2$.

Our method also has computational advantages over approaches that rely on inverting the Hessian. Firstly, implicitly inverting the Hessian is a complex and costly process — by finding only the important components of the projection explicitly, our method is simpler and more efficient. Furthermore, we only need to find these components once; we can reuse them for future test inputs. This type of caching is not possible with methods which require us to calculate the inverse Hessian-vector product for each new test input (e.g. conjugate gradient descent).

### 4.2 RELATED WORK IN DETECTING EXTRAPOLATION

Some recent works explore the relationship between test points, the learned model, and the training set. Several papers examine reliability criteria that are based on distance in some space: within/between-group distances (Jiang et al., 2018), a pre-specified kernel in a learned embedding space (Card et al., 2019), or the activation space of a neural network (Papernot & McDaniel, 2018). We implement some nearest-neighbor baselines inspired by this work in Sec. 5. Additionally, a range of methods exist for related tasks, such as OOD detection (Choi & Jang, 2018; Liang et al., 2017; Gal & Ghahramani, 2015; Schölkopf et al., 2001) and calibration (Naeini et al., 2015; Guo et al., 2017). Some work using generative models for OOD detection uses related second-order analysis (Nalisnick et al., 2018).

A line of work explores the benefits of training ensemble methods explicitly, discussed in detail in Dietterich (2000). These methods have been discussed for usage in some of the applications we present in Sec. 5, including uncertainty detection (Lakshminarayanan et al., 2017), active learning (Melville & Mooney, 2004) and OOD detection (Choi & Jang, 2018). A number of approximate

ensembling methods have also been proposed. For example, MC-Dropout (Gal & Ghahramani, 2015) and Bayes by Backprop (Blundell et al., 2015) are approximate Bayesian model averaging methods that leverage special training procedures to represent an ensemble of prediction functions. These target a distinct notion of uncertainty from our loss-preserving ensembles (see Appendix F). Finally, a line of work on "Rashomon sets" explores loss-preserving ensembles more formally in simpler model classes (Semenova & Rudin, 2019; Fisher et al., 2018).

## 5  EXPERIMENTS

In this section, we give evidence that local ensembles can detect extrapolation due to underdetermination in trained models. In order to explicitly evaluate underdetermination, we present a range of experiments where a pre-trained model has a "blind spot", and evaluate its ability to detect when an input is in that blind spot. We probe our method's ability to detect a range of extrapolation, exploring cases where the blind spot is: **1.** easily visualized, **2.** well-defined by the feature distribution, **3.** well-defined by a latent distribution, and **4.** unknown, but where we can evaluate our model's detection performance through an auxiliary task. See Appendix D for experimental details. Code for running the local ensembles method can be found at `https://github.com/dmadras/local-ensembles`.

### 5.1  VISUALIZING EXTRAPOLATION DETECTION

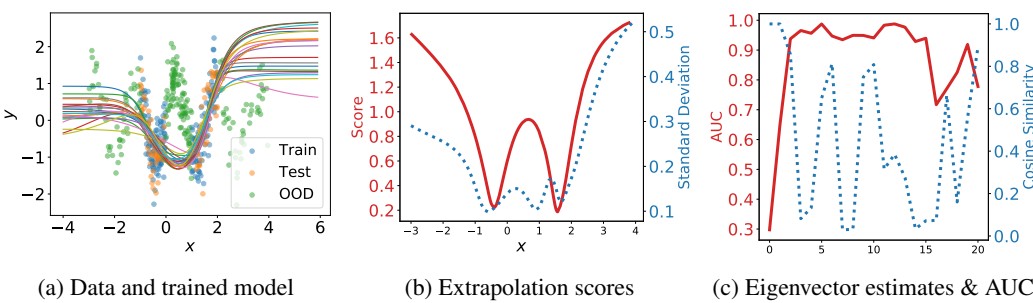

|  (a) Data and trained model | (b) Extrapolation scores | (c) Eigenvector estimates & AUC |

Figure 3: We train a neural network ensemble (Fig. 3a). We compute extrapolation scores (solid line), which correlate with the standard deviation of the ensemble (dotted line) (Fig. 3b). Our OOD performance achieves high AUC (solid line) even though some of our eigenvector estimates have low cosine similarity to ground truth (dotted line) (Fig. 3c).

We begin with an easily visualized toy experiment. In Fig. 3a, we show our data ($y = \sin 4x + \mathcal{N}(0, \frac{1}{4})$). We generate in-distribution training data from $x \in [-1, 0] \cup [1, 2]$, but at test time, we consider all $x \in [-3, 4]$. We train 20 neural networks with the same architecture (two hidden layers of three units each). As shown in Fig. 3a, the ensemble disagrees most on $x < -1, x > 2$. This means that we should most mistrust predictions from this model class on these extreme values, since there are many models within the class that perform equally well on the training data, but differ greatly on those inputs. We should also mistrust predictions from $x \in [0, 1]$, although detecting this extrapolation may be harder since the ensemble agrees more strongly on these points.

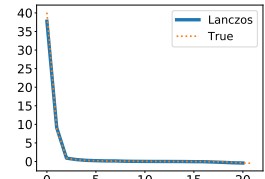

Figure 4: True and estimated eigenspectrums for toy model Hessian. We note that the first few eigenvalues account for most of the variation, and that our estimates are accurate.

For each model in the ensemble, we test our method on an OOD task: can we flag test points which fall outside the training distribution? Since OOD examples may be uncommon in practice, we use AUC to measure our performance. We show that the extrapolation score is empirically related to the standard deviation of the ensemble's predictions at the input, which in turn is related to whether the input is OOD (Fig. 3b). Examining one model from this ensemble, we observe that by estimating only $m = 2$ eigenvectors, we achieve $> 90\%$ AUC (Fig. 3c). It turns out that $m = 10$ performs best on this task/model. As we complete more iterations ($m > 10$) we start finding smaller eigenvalues, which are more important to the ensemble subspace and whose eigenvector we do not wish to project out.

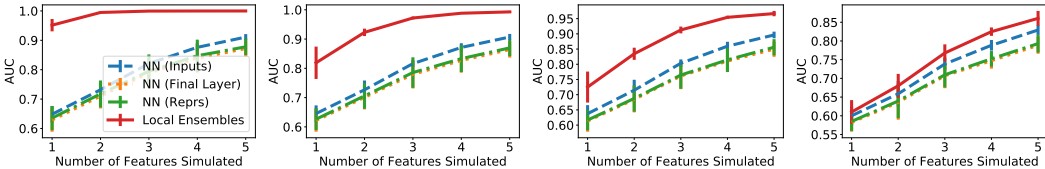

Figure 5: AUC achieved on WineQuality simulated features task (y-axis) compared to the number of extra features simulated (x-axis). Noise parameter $\sigma$ increased from left to right $\sigma \in \{0, 0.1, 0.2, 0.5\}$. Solid line is our method. Results averaged across 5 random seeds, standard deviations shown. Results for the other three datasets are qualitatively similar and shown in Appendix C.

We note our AUC improves even with some eigenvector estimates having low cosine similarity to the ground truth eigenvectors (the Lanczos iteration has some stochasticity due to minibatch estimation — see Appendix B for details). We hypothesize this robustness is because the ensemble subspace of this model class is relatively low-dimensional. Even if an estimated vector is noisy, the non-overlapping parts of the projection will likely be mostly perpendicular to the ensemble subspace, due to the properties of high-dimensional space.

## 5.2 SIMULATED FEATURES

In this experiment, we create a blind spot in a given dataset by extending and manipulating the data's feature distribution. We induce a collinearity in feature space by generating new features which are a linear combination of two other randomly selected features in the training data. This means there is potential for underdetermination: multiple, equally good learnable relationships exist between those features and the target. However, at test time, we will sometimes sample these simulated features from their *marginal* distribution instead. This breaks the linear dependence, requiring extrapolation (the model is by definition underconstrained), without making the new data trivially out-of-distribution. We can make this extrapolation detection task easier by generating several features this way, or make it harder by adding some noise $\sim \mathcal{N}(0, \sigma^2)$ to these features.

We run this experiment on four tabular datasets, training a two hidden-layer neural network for each task. We compare to three nearest-neighbour baselines, where the metric is the distance (in some space) of the test point to its nearest neighbour by Euclidean distance in an in-distribution validation set. *NN (Inputs)* uses input space; *NN (Reprs)* uses hidden representation space, which is formed by concatenating all the activations of the network together (inspired by Papernot & McDaniel (2018), who propose a similar method for adversarial robustness); and *NN (Final Layer)*, uses just the final hidden layer of representations. We note that since our method is *post-hoc* and can be applied to any twice-differentiable pre-trained model, we do not compare to training-based methods e.g. those producing Bayesian predictive distributions (Gal & Ghahramani, 2016; Blundell et al., 2015). Our metric is AUC: we aim to assign higher scores to inputs which break the collinearity (where the feature is drawn from the marginal), than those which do not. In Figure 5, we show that local ensembles (LE) outperform the baselines for each number of extra simulated features, and that this performance is fairly robust to added noise.

## 5.3 CORRELATED LATENT FACTORS

Here, we extend the experiment from Section 5.2 by inducing a blind spot in *latent* space. The rationale is similar: if two latent factors are strongly correlated at training time, the model may grow reliant on that correlation, and at test-time may be underdetermined if that correlation is broken.

We use the CelebA dataset (Liu et al., 2015) of celebrity faces, which annotates every image with 40 latent binary attributes describing the image (e.g. "brown hair"). To induce the blind spot, we choose two attributes: a *label* $L$ and a *spurious correlate* $C$. We then create a training set where $L$ and $C$ are perfectly correlated: a point is only included in the training set if $L = C$. We train a convolutional neural network (CNN), with two convolutional layers and a fully-connected layer at the end, as a binary classifier to predict $L$. Then, we create a test set of held-out data where $P(L = C) = P(L \neq C)$. The test data where $L \neq C$ is in our model's blind spot; these are the inputs for which we want to output high extrapolation scores. We show in Appendix E that the models

dramatically fail to classify these inputs ($L \neq C$). We compare to four baseline extrapolation scores: the three nearest-neighbour methods described in Sec. 5.2, as well as *MaxProb*, where we use $1-$ the maximum outputted probability of the softmax. We test two values of $L$ (*Male* and *Attractive*) and two values of $C$ (*Eyeglasses* and *WearingHat*). We chose these specific values of $L$ because they are difficult to predict and holistic i.e.and not localized to particular areas of image space. In Table 2, we present results for each of the four $L, C$ settings, showing both the loss gradient and the prediction gradient variant of local ensembles. Note that the loss gradient cannot be calculated at test time since we do not have labels available — instead, we calculate a separate extrapolation score using the gradient for the loss with respect to each possible label, and take the minimum. Our method achieves the best performance on most settings, and is competitive with the best baseline on each. However, the variation between the tasks is quite noteworthy. We note two patterns in particular. Firstly, we note the performance of *MaxProb* and the loss gradient variant of our method are quite correlated, which we hypothesize relates to $\nabla_{\hat{Y}}\ell$. Additionally, the effect of increasing $m$ is inconsistent between experiments: we discuss possible relationships to the eigenspectrum of the trained models. See Appendix E for a discussion on these patterns.

## 5.4 ACTIVE LEARNING

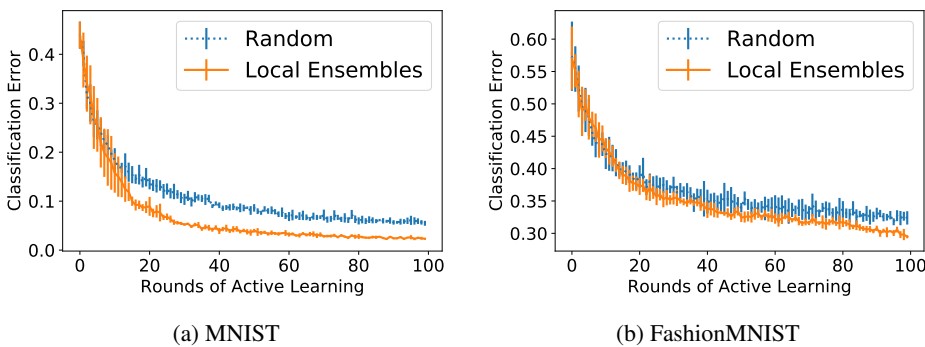

(a) MNIST          (b) FashionMNIST

Figure 6: Active learning results. Average of 5 random seeds shown with std. deviation error bars.

Finally, we consider the case where we know our model has blind spots, but do not know where they are. We use active learning to probe this situation. If underdetermination is an important factor in extrapolation, then local ensembles should be useful for selecting the most useful points to add to our training set. We use MNIST (LeCun et al., 2010) and FashionMNIST (Xiao et al., 2017) for our active learning experiments. We begin the first round with a training set of twenty: two labelled data points from each of the ten classes. In each round, we train our model (a small CNN with two convolutional layers and a fully connected layer at the end) to a mini-

| Method | M/E | M/H | A/E | A/H |
|---|---|---|---|---|
| MaxProb | 0.738 | **0.677** | 0.461 | 0.433 |
| NN (Pixels) | 0.561 | 0.550 | **0.521** | 0.547 |
| NN (Reprs) | 0.584 | 0.578 | **0.503** | 0.533 |
| NN (Final Layer) | 0.589 | 0.517 | 0.480 | 0.497 |
| LE (Loss) | **0.770** | **0.684** | 0.454 | 0.456 |
| LE (Predictions) | 0.364 | 0.544 | **0.519** | **0.582** |

Table 2: AUC for Latent Factors OOD detection task. Column heading denotes in-distribution definitions: labels are $M$ (Male) and $A$ (Attractive); spurious correlates are $E$ (Eyeglasses) and $H$ (Wearing Hat). Image is in-distribution iff label = spurious correlate. LE stands for local ensembles. Each Lanczos iteration uses 3000 eigenvectors. 500 examples from each test set are used. 95% CI is bolded.

mum validation loss using the current training set. After each round, we select ten new points from a randomly selected pool of 500 unlabelled points, and add those ten points and their labels to our training set. We compare local ensembles (selecting the points with the highest extrapolation scores using the loss-gradient variant) to a random baseline selection mechanism. In Fig. 6, we show that our method outperforms the baseline on both datasets, and this improvement increases in later rounds of active learning. We only used 10 eigenvectors in our Lanczos approximation, which we found to be a surprisingly effective approximation; we did not observe improvement with more eigenvectors. This

experiment serves to emphasize the flexibility of our method: by detecting an underlying property of the model, we can use the method for a range of tasks (active learning as well as OOD detection).

# 6 CONCLUSION

We present *local ensembles*, a post-hoc method for detecting extrapolation due to underdetermination in a trained model. Our method uses local second-order information to approximate the variance of an ensemble. We give a tractable implementation using the Lanczos iteration to estimate the largest eigenvectors of the Hessian, and demonstrate its practical flexibility and utility. Although this method is not a full replacement for ensemble methods, which can characterize more complexity (e.g. multiple modes), we believe it fills an important role in isolating one component of prediction unreliability. In future work, we hope to scale these methods to larger models and to further explore the properties of different stopping points $m$. We also hope to explore applications in fairness and interpretability, where understanding model and training bias is of paramount importance.

# 7 ACKNOWLEDGEMENTS

Thank you to Jamie Smith, Yaniv Ovadia, Yoni Halpern, Pang Wei Koh, Jackson Wang, James Lucas, Marc-Etienne Brunet, Kamyar Ghasemipour and Elliot Creager for helpful comments and discussions.

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

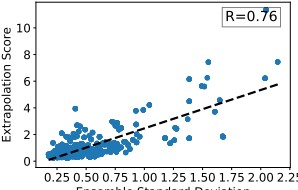 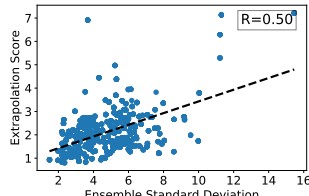 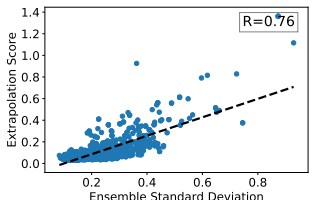

Figure 7: Datasets are (left to right) Boston, Diabetes, Abalone. Dotted line represents linear fit of data. R is Pearson correlation coefficient.

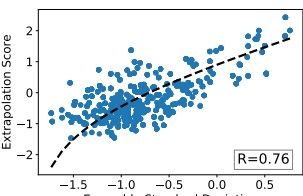 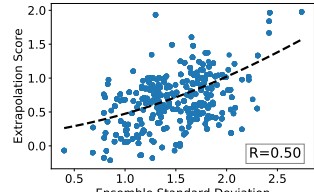 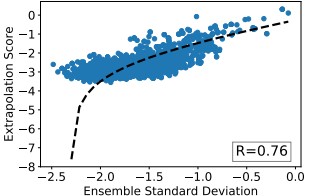

Figure 8: Datasets are (left to right) Boston, Diabetes, Abalone. Each axis is log-scaled. Dotted line represents linear fit of data. R is Pearson correlation coefficient.

## A    ENSEMBLES: OTHER DATASETS

We show in Figures 7 and 8 the same plots as Fig. 2 for other tabular datasets, demonstrating the strongly linear relationship between extrapolation score and ensemble standard deviation.

## B    THE LANCZOS ITERATION: FUTHER DETAILS

The Lanczos iteration (Lanczos, 1950) is a method for tridiagonalizing a Hermitian matrix. It can be thought of as a variant of power iteration, iteratively building a larger basis through repeatedly multiplying an initial vector by $M$, ensuring orthogonality at each step. Once $M$ has been tridiagonalized, computing the final eigendecomposition is relatively simple — a number of specialized algorithms exist which are relatively fast ($O(p^2)$) (Dhillon, 1997; Cuppen, 1980).

The Lanczos iteration is simple to implement, but presents some challenges. The first challenge is numerical instability — when computed in floating point arithmetic, the algorithm is no longer quarnteed to return a good approximation to the true eigenbasis, or even an orthogonal one. As such, the standard implementation of the Lanczos iteration is unstable, and can be inaccurate even on simple problems. Fortunately, solutions exist: a procedure known as two-step classical Gram-Schmidt orthogonalization — which involves ensuring *twice* that each new vector is linearly independent of the previous ones — is guaranteed to produce an orthogonal basis, with errors on the order of machine roundoff (Giraud et al., 2005). A second potential challenge is presented by the stochasticity of minibatch computation. Since we access our Hessian $H$ only through Hessian-vector products, we must use only minibatch computation at each stage. This means that each iteration will be stochastic, which will decrease the accuracy (but not the orthogonality) of the eigenvector estimates provided. However, in practice, we found that even fairly noisy estimates were nonetheless useful — see Sec. 5.1 for more discussion.

### B.1    LANCZOS ALGORITHM CODE SNIPPET

The Lanczos algorithm is quite simple to implement. Figure 9 shows a a short implementation using Python/Numpy (Oliphant, 2006).

```
import numpy as np

def lanczos(matmul_fn, dim, num_iters, eps=1e-8):
    '''Given implicit access to a matrix M of size (dim x dim)
    through mamtul_fn, tridiagonalize M into diagonal elements
    Alpha and off-diagonal elements Beta. Also return the
    associated orthonormal basis Q.'''

    # Initialize orthonormal basis.
    Q = [np.zeros((dim, 1))]
    # Initialize off-diagonal elements.
    Beta = []
    # Initialize diagonal elements.
    Alpha = []

    # Begin with random initial vector.
    q = np.random.uniform(size=(dim, 1))
    Q.append(q / np.linalg.norm(q))
    Q_k_range = Q[0]

    for k in range(1, num_iters + 1):
        # Compute next step of power iteration.
        z = matmul_fn(Q[k])
        Alpha.append(np.matmul(Q[k].T, z))
        Q_k_range = np.concatenate([Q_k_range, Q[k]], axis=1)

        # Reorthogonalize using two-step classical Gram-Schmidt.
        for _ in range(2):
            z_orth = np.sum(np.matmul(z.T, Q_k_range) *
                    Q_k_range, axis=1, keepdims=True)
            z = z - z_orth

        Beta.append(np.linalg.norm(z))
        Q.append(z / Beta[k])
        # Check for convergence.
        if np.linalg.norm(z) < eps:
            break

    return Q, Beta, Alpha
```

Figure 9: Example Python implementation of Lanczos algorithm for tridiagonalizing an implicit matrix $M$.

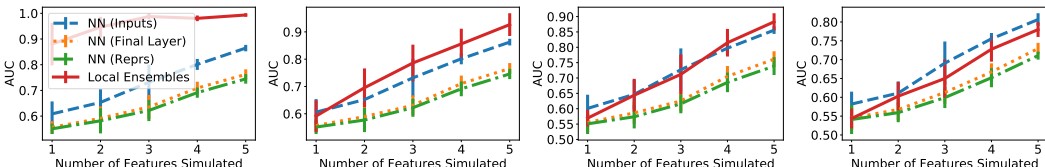

Figure 10: Boston dataset

## C  SIMULATED FEATURES - OTHER DATASETS

In Section 5.2, we present results for an experiment where we aim to detect broken collinearities in the feature space. In Figures 10, 11, and 12, we show results on three more tabular datasets. See the main text for more experimental details.

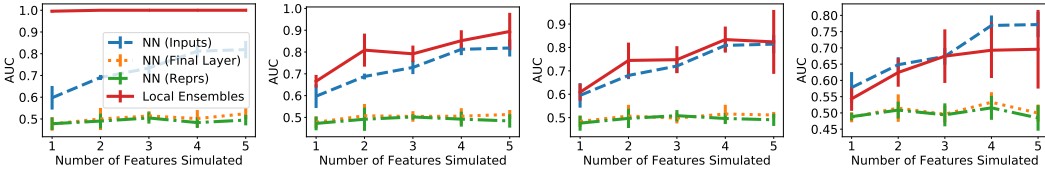

Figure 11: Diabetes dataset

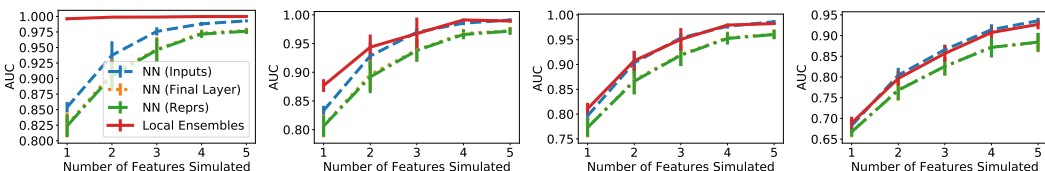

Figure 12: Abalone dataset

## D   EXPERIMENTAL DETAILS

Here we give more experimental details on the datasets and models used.

### D.1   DATASETS

#### D.1.1   TABULAR DATASETS

We subtracted the mean and divided by the standard deviation for each feature (as calculated from the training set).

**Boston (Harrison Jr & Rubinfeld, 1978) and Diabetes (Efron et al., 2004).**   These datasets were loaded from Scikit-Learn (Pedregosa et al., 2011).

**Abalone (Nash et al., 1994).**   This dataset was downloaded from the UCI repository (Dua & Graff, 2017) at `http://archive.ics.uci.edu/ml/datasets/Abalone`. We converted sex to a three-dimensional one-hot vector (*M, F, I*).

**WineQuality (Cortez et al., 2009).**   This dataset was downloaded from the UCI repository (Dua & Graff, 2017) at `http://archive.ics.uci.edu/ml/datasets/Wine+Quality`. We used both red and white wines.

### D.2   MNIST, FASHIONMNIST AND CELEBA.

These datasets were loaded using Tensorflow Datasets `https://github.com/tensorflow/datasets`. We divided the pixel values by 255 and, for MNIST and FashionMNIST, binarized by thresholding at 0.7.

### D.3   EXPERIMENTAL DETAILS

#### D.3.1   TOY DATA EXPERIMENTS

For the first experiment, we train a two-layer neural network with 3 hidden units in each layer and tanh units. We train for 400 optimization steps using minibatch size 32. Our data is generated from $y = \sin(4x) + \mathcal{N}(0, \frac{1}{4})$. We generate 200 training points, 100 test points, and 200 OOD points. We aggregate $Y$ over a grid of 10 points from -1 to 1, with aggregation function $\min$. We run the Lanczos algorithm until convergence.

For the second experiment, we train a two-layer neural network with 5 hidden units in each layer and ReLU units. Our data is generated from $y = \beta x^2 + \mathcal{N}(0, 1)$. Our training set consists of $x$ drawn

| Test Set | M/E | M/H | A/E | A/H |
|---|---|---|---|---|
| In-Distribution | 0.03 | 0.06 | 0.01 | 0.02 |
| Out-of-Distribution | 0.98 | 0.96 | 0.90 | 0.93 |

Table 3: Error rate for in and out of distribution test set with correlated latent factors setup. Column heading denotes in-distribution definitions: labels are $M$ (Male) and $A$ (Attractive); spurious correlate are $E$ (Eyeglasses) and $H$ (Wearing Hat). Image is in-distribution iff label == spurious correlate.

uniformly from $[-0.5, 0.5]$ and $[2.5, 3.5]$. However, at test time, we will consider $x \in [-3, 6]$. We generate 200 training points, 100 test points, and 200 OOD points. We aggregate $Y$ over a grid of 5 points from -6 to 9, with aggregation function $\min$.

### D.3.2 SIMULATED FEATURES

For each dataset we use the same setup. We use a two-layer MLP with ReLU activations and hidden layer sizes of 20 and 100. We trained all models with mean squared error loss. We use batch size 64, patience 100 and a 100-step running average window for estimating current performance. For the Lanczos iteration, we run up to 2000 iterations. We always report numbers from the final iteration run. For estimating the Hessian in the HVPs in the Lanczos iteration, we use batch size 32 and sample 5 minibatches.

To pre-process the data, we first split randomly out 30% of the dataset as OOD. We choose 2 random features $i, j$ and a number $\beta \sim \mathcal{U}(0, 1)$, and generate the new feature $\tilde{x} = \beta x[i] + (1 - \beta)x[j]$. We also normalize this feature by its mean and standard deviation as calculated on the training set. We add random noise to the features after splitting into in-distribution and OOD — meaning we are not redrawing from the same marginal noise distribution. We use 1000 examples from in-distribution and OOD for testing.

### D.3.3 CORRELATED LATENT FACTORS

We use a CNN with ReLU hidden activations. We use two convolutional layers with 50 and 20 filters each and stride size 2 for a total of 1.37 million parameters. We trained all models with cross entropy loss. We use an extra dense layer on top with 30 units. We use batch size 32, patience 100 steps, and a 100-step running average window for estimating current performance. We sample the validation set randomly as 20% of the training set. For the Lanczos iteration, we run 3000 iterations. We always report numbers from the final iteration run. We use 500 examples from in-distribution and OOD for testing. For estimating the Hessian in the HVPs in the Lanczos iteration, we use batch size 16 and sample 5 minibatches.

### D.3.4 ACTIVE LEARNING

We use a CNN with ReLU hidden activations. We use two convolutional layers with 16 and 32 layers, stride size 5, and a dense layer on top with 64 units. We trained all models with mean squared error loss. We use batch size 32, patient 100 steps, and a 100-step running average window for estimating current performance.

## E CORRELATED LATENT FACTORS

### E.1 PERFORMANCE OF BINARY CLASSIFIERS

In Section 5.3, we discuss an experiment where we correlated a latent label $L$ and confounder $C$ attribute. Table 3 shows the in-distribution and out-of-distribution test error. These are drastically different, meaning that learning to detect this type of extrapolation is critical to maintain model performance.

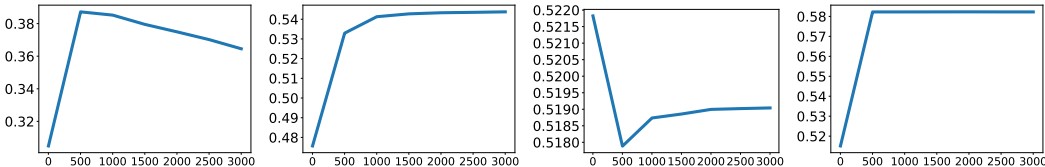

Figure 13: Latent Factors OOD detection task, using loss gradient. Y-axis shows AUC, X-axis shows the number of eigenvectors estimated by the Lanczos algorithm, data sampled every 500 eigenvectors. Tasks from left to right are *M/E*, *M/H*, *A/E*, *A/H*.

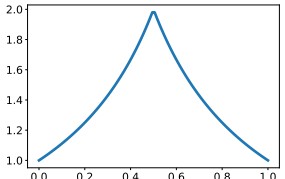

Figure 14: Latent Factors OOD detection task, using loss gradient. Y-axis shows AUC, X-axis shows the number of eigenvectors estimated by the Lanczos algorithm, data sampled every 500 eigenvectors. Tasks from left to right are *M/E*, *M/H*, *A/E*, *A/H*.

### E.2 BEHAVIOUR OF AUC WITH MORE ESTIMATED EIGENVECTORS

In Fig. 13 and 14, we show that the tasks present differing behaviours as more eigenvectors are estimated. We observe that for the *Male/Eyeglasses* and *Attractive/WearingHat* tasks, we get improved performance with more eigenvectors, but for the others we do not necessarily see the same improvements. Interestingly, this upward trend occurs both times that our method achieves a statistically significant improvement over baselines. It is unclear why this occurs for some settings of the task and not others, but we hypothesize that this is a sign that the method is working more correctly in these settings.

### E.3 RELATIONSHIP BETWEEN LOSS GRADIENT AND *MaxProb* METHOD

As discussed in Sec. 5.3, we have the relationship between the loss $\ell$, prediction $\hat{Y}$, and parameters $\theta$: $\nabla_\theta \ell = \nabla_{\hat{Y}} \ell \cdot \nabla_\theta \hat{Y}$. Using $\min$ as an aggregation function, we find that $\min_{Y \in \{0,1\}} |\nabla_{\hat{Y}} \ell(Y, \hat{Y})|$ has an inverted V-shape (Fig. 15). This is a similar shape to $1 - \max(\hat{Y}, 1 - \hat{Y})$, which is the metric implicitly measured by *MaxProb*.

### E.4 ESTIMATED EIGENSPECTRUM OF DIFFERENT CORRELATED LATENT FACTOR TASKS

In Fig. 16, we examine the estimated eigenspectrums of the four tasks we present in the correlated latent factors experiment, to see if we can detect a reason why performance might differ on these tasks.

In Fig. 16a, we show that the two tasks with the label *Attractive*, the eigenvalues are larger. These are also the tasks where the loss-gradient-based variant of local ensembles failed, indicating that that variant of the method may be worse at handling larger eigenvalues.

Figure 15: X-axis is $\hat{Y}$, Y-axis is $\min_{Y \in \{0,1\}} \nabla_{\hat{Y}} \ell(Y, \hat{Y})$.

In Fig. 16b, we show that the two tasks where the local ensembles methods performed best ($M/E$, $A/H$, achieving statistically significant improvements over the baselines at a 95% confidence level, and also showing improvement as more eigenvectors were estimated), the most prominent negative eigenvalue is relatively smaller magnitude compared to the most prominent positive eigenvalue. This could mean that the local ensembles method was less successful in the other tasks ($M/H$, $A/E$)

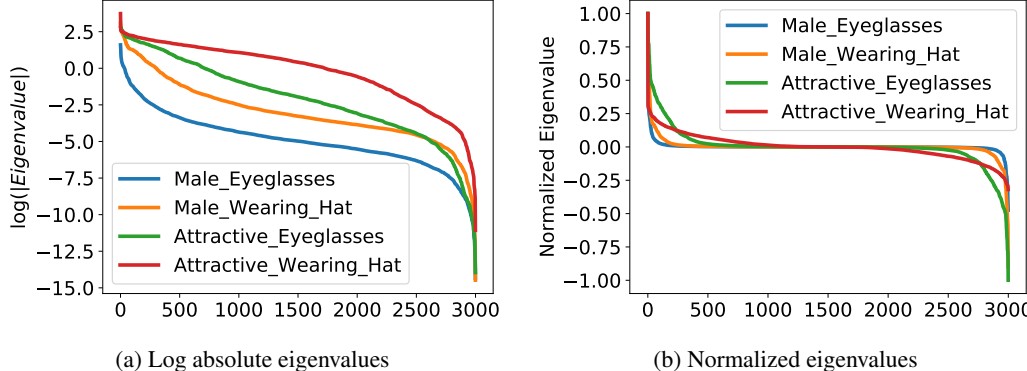

(a) Log absolute eigenvalues        (b) Normalized eigenvalues

Figure 16: We show the estimated eigenspectrum of the four CNNs we train on the correlated latent factors task. In Fig. 16a, we show the absolute estimated eigenvalues sorted by absolute value, on a log scale. In Fig. 16b, we show the estimated eigenvalues divided by the maximum estimated eigenvalues. We only show 3000 estimated eigenvalues because we ran the Lanczos iteration for only 3000 iterations, meaning we did not estimate the rest of the eigenspectrum.

simply because those models were not trained close enough to a convex minimum and still had fairly significant eigenvalues.

## F    EMPIRICAL COMPARISON TO MC-DROPOUT

One of the strengths of our method is that it can be applied *post-hoc*, making it usable in a wide range of situations. Therefore, we choose to compare the local ensembles method to only baselines which can also be applied post-hoc. However, one might still be interested in how some of these methods compare against ours. One such method is the MC-Dropout method (Gal & Ghahramani, 2016), which constructs an ensemble at test time from a model trained with dropout, by averaging over several random dropout inferences. This method is not considered post-hoc, since it is only applicable to methods which have been trained with dropout (and works best for models with dropout on each layer, rather than just a single layer).

Both MC-dropout and local ensembles estimate the variance of an ensemble achieved through perturbations of a trianed model. In MC-dropout, those perturbations take the form of stochastically setting some weights of the model to 0; on local ensembles, they are Gaussian perturbation projected into the ensemble subspace. However, despite these similarities, we provide empirical evidence here that our method is categorically different than MC-Dropout, by showing that our perturbations find a "flatter" (more loss-preserving) ensemble. This means that our method is more directly measuring underdetermination, since we are comparing nearby models which are equally justified by the training data.

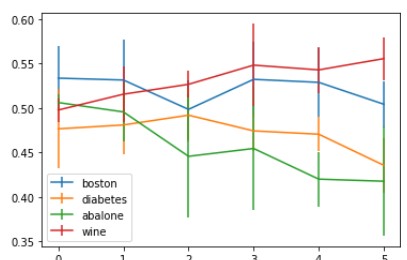

Figure 17: Performance of MC-dropout on the simulated features task from Sec. 5.2. Y-axis is AUC, X-axis is number of features simulated. This is the 0 noise setting.

We use the four tabular datasets discussed in Sec 5.2. For each dataset, we train a neural network with dropout on each weight (as in MC-dropout), to minimize a mean squared error objective. We use a dropout parameter $p = 0.1$, as suggested in Gal & Ghahramani (2016). We then create an MC-dropout ensemble of size 50: that is, we randomly dropout each weight with probability 0.1, and repeat 50 times. We calculate the training loss for each model in the MC-dropout ensemble. To estimate how "flat" this ensemble is (i.e. how loss-preserving on the training set), we simply subtract the training loss of the full model (no weights dropped out) from each model in the dropout-

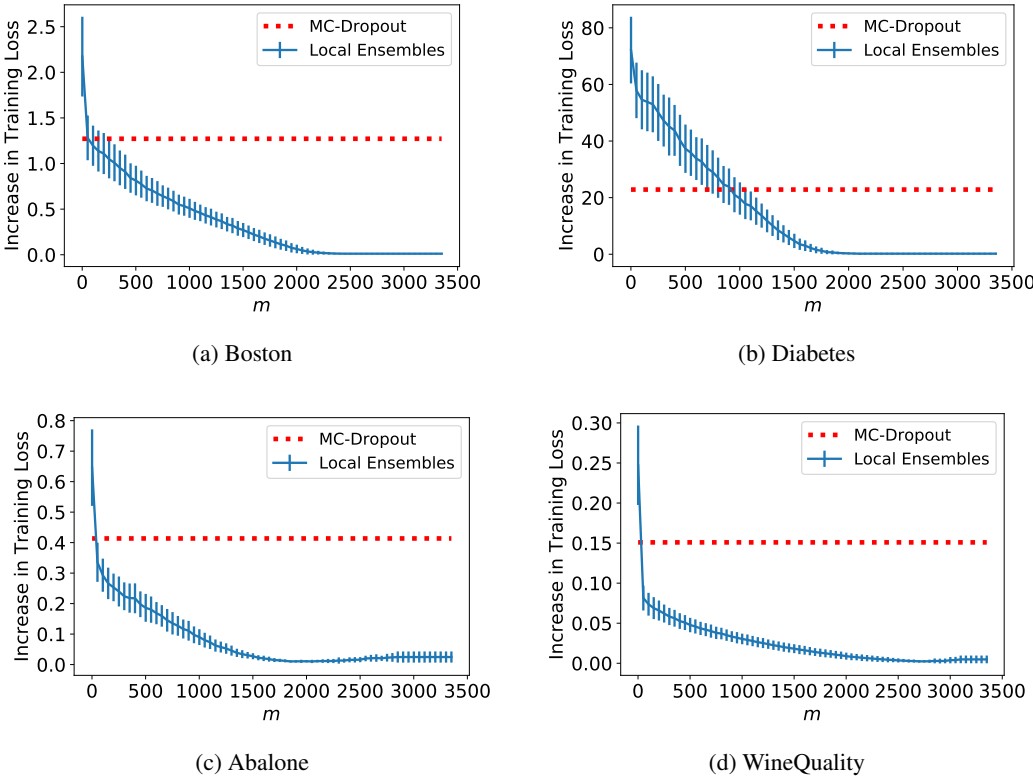

(a) Boston

(b) Diabetes

(c) Abalone

(d) WineQuality

Figure 18: Comparison of loss-preservingness of ensembles found by MC-Dropout vs Local Ensembles methods. X-axis shows different values for $m$ (number of dimensions projected out), with $p - m$ dimensions in the ensemble subspace. Y-axis shows expected increase in training loss across the ensemble, as compared to the non-perturbed model (no weights dropped out/perturbation = 0). Gaussian perturbation is scaled to have the same expected size as dropout perturbations for that task and $m$-value. Standard deviations shown across 20 random initializations. For each initialization, 50 perturbations were performed. Dotted red horizontal line shows average training loss increase for an MC-dropout ensemble (20 models, 50 perturbations each).

ensemble, and average these differences. We repeat this process 20 times, changing only the random seeds.

We can now estimate a similar quantity (average increase in training loss across the ensemble) for our method. For each dataset, we train a neural network with the same architecture as in the dropout experiment. We then choose a standard deviation $\sigma$ equal to the average magnitude of the perturbations created through MC-dropout on the corresponding task. This is to ensure a fair comparison — for each method, we are using perturbations on the same scale. We run the Lanczos iteration to convergence. We then compute the ensemble subspace for each Lanczos iteration $m$ (recall that $p - m$ is the dimensionality of the ensemble subspace, and $m$ is the number of dimensions projected out). For each ensemble subspace of dimensionality $p - m$, we sample Gaussian noise from $\mathcal{N}(0, \frac{\sigma}{\sqrt{p-m}})$, project it into the ensemble subspace, and add it to the model's trained parameters. This ensures that when projected into the ensemble subspace, the Gaussian perturbation has the same expected magnitude as the dropout perturbations for the corresponding task. Using these projected Gaussian perturbations, we create an ensemble of size 50 for each $m$, and calculate the training loss for each model in the local ensemble. We subtract the training loss of the original model (perturbation 0) from each model in the local ensemble, and average across the ensemble. We repeat this process 20 times, changing only the random seeds.

In Figure 18, we show the results of this comparison for each of the four datasets. The decreasing curve shows that as we project out more eigenvectors (increasing $m$), the resulting ensemble subspace becomes flatter, i.e. perturbations in this subspace increase training loss by a smaller amount. The

horizontal dotted line is the average training loss increase for model in an an MC-dropout ensemble. We see that this horizontal line is well above the descending curve representing local ensembles. Recall that we scaled the perturbations to be the same size for both MC-Dropout and all value of $m$ for local ensembles. This means that, for perturbations of the same magnitude, for most values of $m$, the ensembles found by our method are much flatter than those found by MC-dropout. While MC-dropout ensembles may be useful for other tasks, this shows that our method is more directly measuring underdetermination in the model. Additionally, we run the MC dropout method on the simulated features task from Sec. 5.2. We find (see Fig. 17) that it is unable to match the performance of our method, reinforcing the point that it is concerned mostly with other types of uncertainty.

