# OpenReview forum: "Detecting Extrapolation with Local Ensembles"
_ICLR.cc/2020/Conference — Accept (Poster)_

### Official Review · AnonReviewer3 · 2019-10-21
**Official Blind Review #3**

**Rating:** 6

**Review:**

# Summary of contribution
- The paper provides a novel fast and simple approximation of second-order local parameter sensitivity of neural networks, to estimate a form of uncertainty wrt to a test sample, which is further used and tested as a novelty detector.
- The method analyzes the most significant eigenvector/eigenvalues of the Hessian (of training loss), and use the compliment of their span to get directions of local perturbations to network parameters that affect training loss little ("ensemble subspace"). The novelty score is then based on how much the prediction is influenced by these perturbations.
- The idea of estimating "ensemble subspace" is interesting and computationally effective. Compared to other recent methods that also use second-order gradients for uncertainty, this paper is more generally applicable, and can be faster at test time. The paper demonstrates good performance on both simulated data and real data (CelebA faces with CNN, etc.).

# Decision TL;DR
I am giving a weak reject. The paper is strong in its idea, formulation, and theory, but is too similar to recent related works which this paper is reluctant to compare to (either in theory, efficiency, or performance). Since the contribution of the paper lies in the efficient approximation of local ensemble methods, readers cannot gauge how beneficial the contribution is compared to other approximations of ensembles.


# Pros
- Novel way to estimate a local neighborhood that affects training loss little (Note: not an expert in this line of research, not sure if it is completely novel) by estimating significant eigenvectors and using their compliment space.
- The paper is well-written, and relatively easy to understand, despite a few hard-to-follow spots
- Widely applicable post-hoc to any trained neural networks, and potentially faster training than ensembles / Bayesian approximated ensembles
- (Theoretical) stability compared to full Hessian inversion

# Cons
Motivation wrt other papers unclear, and a lack of comparison.
- Two of the cited papers (Gal & Ghahramani, 2016; Blundell et al., 2015) both work on local ensembles. The former uses MC dropout, the latter estimates a diagonal covariance of a Gaussian distribution of network parameters. These methods are not mentioned in the motivation or related work, which makes it hard to say this paper is well-placed in the literature.
- The reason that these methods are not compared to is insufficient. The paper only argues that they are not "post-hoc" methods. It is very unclear why in any circumstance (or use case) a post-hoc estimation of local ensemble must be (or is preferred to be) used, rather than having network parameters and local neighborhood jointly estimated. If it is for efficiency reasons, the paper does not provide any experimental comparison of the efficiency. Also, it is hard to argue that the "post-hoc" nature of this paper makes it so different from the two prior work. For the first prior work, the only time it is not post-hoc is when the original network does not have any dropout layer, and that circumstance is not very common. For the second prior work, one can easily make it post-hoc by training the network first, and estimate the diagonal covariance post-hoc using their loss.
- The advantage of this paper is that it is more efficient and stable than alternatives, but only  the full hessian inversion is discussed. In particular, it may be necessary to discuss this paper's efficiency against MC-dropout (Gal & Ghahramani, 2016). This method can be done in mini-batches, while the proposed method has to run forward and back-propagation separately for each sample to get gθ*(x'), and it is unclear how well that scales.

Efficiency analysis lacking
- As discussed above, the proposed method seems to need to back-prop for each test sample separately without using a batch. How much this affects test efficiency is unclear.

Experimental comparison with similar methods missing.
- The paper would benefit from comparing to the two cited papers (among which MC-dropout is so easy to implement) as well as a full hessian estimation (for toy datasets at least).
- The paper poses itself as an efficient alternative, so it would be essential to gauge experimentally how fast each method is.

Others. (not crucial issues)
- The performance of the paper's main method (LE w/ predictions) underperforms in Table 2, and a variant had to be proposed to make up for the performance drop. This suggests instability of the proposed method wrt new datasets.
- The claim in contribution "We identify underdetermination as a key factor in the unreliability of predictions" is not verified.
- Inability to scale up to large networks with larger m needed, compared to real ensemble methods or Bayesian networks with Gaussian distributions.


# Room for improvement (decreasing order of importance)
- Improve placement in the literature by discussing when this paper is more useful than prior work (Gal & Ghahramani, 2016; Blundell et al., 2015).
- Detailed theoretical or experimental analysis of efficiency against alternative approximations of ensembles.
- Performance comparison to ensemble and local Bayesian methods.

# Editorial issues
- Figure 3(c) x axis meaning unclear
- Figure 4 not mentioned in text, and unclear which experiment this refers to
- Table 2 experiment's loss gradient version is not explained.


#############################################################
POST REBUTTAL
#############################################################

TL;DR: The rebuttal addresses some but not all of my concerns. The MC-dropout comparison especially shows the difference between some approximate ensemble methods and ensembles that specifically changes the loss little (this paper). In the end, it is a good paper in terms of theory, although the experiments is lacking in crucial places (no comparison with many existing papers that do attempt to invert hessian) and lack of analysis of claimed efficiency, which I can only hope don’t turn out to be a big deal. I am increasing the score, to marginally above borderline, but please consider the following feedback for the camera-ready version.

> MC Dropout and Bayes by Backprop effectively measure different types of uncertainty from what our method targets, and so they should not be considered competing methods.
I disagree; by the same logic we can never compare SVM with random forest because they are so different.

> However, as we discussed in 4.1, small eigenvalues make turning this representation into a proper posterior distribution difficult, because these eigenvalues need to be inverted to obtain a covariance matrix, but these inverted eigenvalues can be numerically infinite.
I agree; but the main difference is the full hessian is hard to implement while there are papers that have implemented the diagonal hessian. It would benefit the paper to compare to that and prove the issue of instability on top of arguing theoretically.

> First, we clarify that our method can in fact be performed using mini-batches of test points (both the forward and backward passes).
As far as I know, with mini-batches, the gradients averaged over all samples in the mini-batch are computed for each network parameter. But this method needs the gradient wrt each sample separately.
If the authors have some implementation trick that allows computing network parameters’ gradients wrt each sample in the mini-batch separately, please include in the implementation details. Otherwise, please mention this drawback in any “fast” claim.

I will change the score upwards due to the informative rebuttal. Please include much of the discussion in the paper or appendix.


**Experience Assessment:**

I have published one or two papers in this area.

**Review Assessment: Checking Correctness Of Derivations And Theory:**

I carefully checked the derivations and theory.

**Review Assessment: Checking Correctness Of Experiments:**

I carefully checked the experiments.

**Review Assessment: Thoroughness In Paper Reading:**

I read the paper at least twice and used my best judgement in assessing the paper.

---

> ### Author Response · Authors · 2019-11-14
> **Response to Reviewer 3 (part 1)**
>
> We thank the reviewer for their thorough engagement with our paper. The points raised here helped us better contextualize our work in the existing literature, and motivated a number of experiments that we found to be enlightening. We address specific points below.
>
> # Major points
>
> > “[MC Dropout (Gal & Ghahramani, 2016) and Bayes by Backprop (Blundell et al., 2015)] are not mentioned in the motivation or related work, which makes it hard to say this paper is well-placed in the literature.”
>
> We appreciate the reviewer’s concern that we do not spend as much space in the paper contrasting against these two existing approximate ensemble-like methods. We focused mostly on second-order approximate uncertainty quantification methods (Section 4.1) because our method resembles these most closely. However, the requested comparison is also useful.
>
> Our general thoughts here are that MC Dropout and Bayes by Backprop effectively measure different types of uncertainty from what our method targets, and so they should not be considered competing methods. We have clarified this position in the paper, in both the introduction and the related work sections. We have also added an appendix that demonstrates this qualitative distinction empirically.
>
> MC Dropout constructs an ensemble of predictors by performing a set of stochastic forward passes through the network by applying dropout at test time. MC Dropout is expected to work well when the network was trained with dropout. In particular, Gal and Ghahramani show that this procedure approximates the posterior predictive distribution of a variational approximation to a deep GP. Similarly, Bayes by Backprop is a method for constructing a variational posterior over model weights. This method requires that the network be trained to minimize a variational ELBO. Thus, as mentioned in the paper, both methods require some assumptions on the training process (more about the post-hoc question later).
>
> Similarly to our method, both of these methods can be framed as constructing a (weighted) ensemble of predictors and quantify predictive uncertainty in terms of the variation of predictions across this ensemble. However, the motivations for these ensembles are quite different from the method that we propose. Specifically, the ensembles constructed under these approximate Bayesian methods have no guarantee of being loss-preserving (that is, the members of the ensemble are not guaranteed to have near-identical loss) because they are not constructed with this being the goal.
>
> We have implemented some experiments with MC Dropout on the small UCI datasets to demonstrate how these targets of estimation differ. These are now included in Appendix F. We implement a dropout scheme, and measure how much the training loss is perturbed across a Dropout ensemble. We then measure how a parameter perturbation with the same norm as the Dropout perturbation affects the training loss when that perturbation is constrained to lie in an ensemble subspace, for many values of our tuning parameter m (larger m corresponds to smaller eigenvalues in the ensemble subspace, and thus a flatter subspace). The figures clearly show that the variation in training loss in the Dropout ensemble is much larger than the variation in training loss within our constructed local ensembles when m is set so that the corresponding eigenvalue \lambda^{(m)} is reasonably small (in the experiments in the main test, we set m = 2000).
>
> This is in line with some of the arguments we make to motivate our method. We can expand on them here. In general, underdetermination is a narrower notion of uncertainty than Bayesian uncertainty — Bayesian uncertainty also incorporates uncertainty from dimensions of the parameter space that are well-constrained by the training data. In principle, exact Bayesian inference with appropriate priors could capture the notion of underdetermination that we aim to capture with our method. However, approximations to Bayesian inference and high-dimensional prior distributions often result in poor representations of underdetermination.

---

> > ### Author Response · Authors · 2019-11-14
> > **Response to Reviewer 3 (part 1b)**
> >
> > For example, variational approximations such as Bayes by Backprop with a variational Gaussian posterior will impose curvature on the loss surface because the approximating distribution is unimodal. Thus, the variational posterior distribution will either artificially introduce curvature to underdetermined dimensions in the parameter space, or it will become nearly improper, and thus numerically unstable. In practice, a user is likely to add strong regularization until evidence of underdetermination has been erased (similarly to fixes to ill-conditioned Hessians discussed in Section 4.1). Likewise, in the case of MC Dropout, the implicit prior distribution that is used to marginalize over weights in the Dropout scheme can result in a posterior distribution that does not represent underdetermination well. In high dimensions, it is often the case the prior distributions will concentrate mass in counterintuitive places, resulting in posterior distributions that do not preserve salient properties of the log-likelihood (i.e., negative loss). In the case of Dropout, there is no guarantee that the posterior distribution will retain the “flat” directions in the loss surface that we are interested in exploring. This is borne out in the experiments cited above.

---

> ### Author Response · Authors · 2019-11-14
> **Response to Reviewer 3 (part 2)**
>
>
> > Advantages of post-hoc methods, and comparison to post-hoc-ness of MC Dropout and Bayes by Backprop.
>
> We would like to push back a bit on the reviewer’s points about the lack of importance of post-hoc methods. Especially for large models, it is quite common in practice to make use of a pre-trained model even if the dataset is available. Often, the person deploying the model does not have full control over the procedure used to train the model, or would prefer not to retrain a performant model from scratch. In these cases, post-hoc methods are appealing.
>
> We do agree that if Dropout happens to be used in network training, that MC Dropout is an effective technique for cheaply generating an ensemble of predictors. However, not all neural networks are trained with Dropout and, as we discussed above, MC Dropout is targeting a very different form of uncertainty. In fact, the reliability score generated in this way would be at the mercy of the choices made by the person who trained the model -- the choice of where to include dropout layers, and the dropout probabilities all change the type of uncertainty being estimated. On the other hand, the interpretation of our method does not depend on these choices; it only depends on the learning problem specification, and choices made by the person deploying the model (i.e., the choice of m).
>
> If we understand correctly, the reviewer also suggests deploying Bayes by Backprop in a post-hoc fashion, by estimating only the diagonal elements of the Hessian and using this as an approximation to the inverse covariance matrix of a posterior distribution. Our method could be seen as a way to fix some of the pathologies of this proposed method. First, we note that this suggestion is quite similar to the Laplace approximation approach discussed in Section 4.1. In fact, the Laplace approximation method is one step more sophisticated than this proposal, because (by the eigendecomposition argument we make) it effectively constructs the basis in which the model Hessian can be represented _exactly_ as a diagonal matrix. However, as we discussed in 4.1, small eigenvalues make turning this representation into a proper posterior distribution difficult, because these eigenvalues need to be inverted to obtain a covariance matrix, but these inverted eigenvalues can be numerically infinite.
>
> Our method effectively patches this approach by noting that is is the alignment of the prediction gradient with these “flat” directions that translates to instability in predictions (see the second-to-last paragraph of 4.1). By settling for measuring the projection of the prediction gradient into the “flat” subspace, we abandon some of the full Bayesian interpretation that could come from the Laplace approximation approach, but retain most of the key information about the underdetermined predictions.

---

> ### Author Response · Authors · 2019-11-14
> **Response to Reviewer 3 (part 3)**
>
> > Efficiency compared to MC Dropout and “[MC Dropout] can be done in mini-batches, while the proposed method has to run forward and back-propagation separately for each sample to get gθ*(x'), and it is unclear how well that scales.”
>
> First, we clarify that our method can in fact be performed using mini-batches of test points (both the forward and backward passes).
>
> The reviewer is correct that MC Dropout is probably more computationally efficient than our proposed method, because it only requires stochastic forward passes through the network. However, as we have argued above, MC Dropout does not measure a comparable notion of uncertainty. From this perspective, our point of comparison is fully retrained ensembles, and our method is computationally cheaper.
>
> # Minor Points
> > The performance of the paper's main method (LE w/ predictions) underperforms in Table 2, and a variant had to be proposed to make up for the performance drop. This suggests instability of the proposed method wrt new datasets.
>
> We agree that the difference in performance on this task is curious, and it does suggest that our method has some failure modes. This proposed task is quite challenging, and may not map neatly to uncertainty due to underdetermination — indeed, the strong performance of MaxProb here suggests otherwise. Some other possible issues are discussed in Appendix E.4.
>
> The variant that we propose here is a natural variant more in line with prior work such as influence functions. It is briefly described at the end of section 2, and in more detail in Appendix E.4.
>
> > The claim in contribution "We identify underdetermination as a key factor in the unreliability of predictions" is not verified.
>
> We agree: previous work cited in the introduction identified this factor. We have removed this claim from our contributions.
>
> > Inability to scale up to large networks with larger m needed, compared to real ensemble methods or Bayesian networks with Gaussian distributions.
>
> We agree this is an issue. Our demonstrations in this paper are very much proofs of concept, and in principle, there is no reason this could not scale to larger networks and larger m. We also note that for many networks, taking m << #parameters may be sufficient, given the highly ill-conditioned hessians in many deep models.

---

### Official Review · AnonReviewer2 · 2019-10-24
**Official Blind Review #2**

**Rating:** 6

**Review:**

The paper focusses on underdetermination as being key to extrapolation. In the case of pretrained models, the model extrapolates on a test input if the prediction at this input is underdetermined or multiple predictions are equally consistent with models characterized by similar architecture and loss functions.
The underdetermination in the case of over-parameterized model classes as in deep neural networks is used here as a way of detecting extrapolation.

The authors define an extrapolation score for trained models on unlabelled test point by measuring variance of predictions across an ensemble selected from local perturbations on trained model parameters that fit the training data well or having similar training loss.

The score approximates the variance of predictions by estimating the norm of the component of the test point’s gradient that aligns well with the low curvature directions of the Hessian, thus providing a tractable quantity in quantifying uncertainty in predictions. The motivation is that if the models have been trained to a local minimum or saddle point, then parameter perturbations in flat directions (small eigenvalues of the Hessian) do not change the training loss substantially.  These models with small perturbations on the flat regions then form the local ensemble for measuring the extrapolation and predicting on out of distribution samples, spurious correlated samples and for  active learning on uncertain data.

The authors prove that the extrapolation score is proportional to the standard deviations of predictions across a model ensemble with similar training loss. The math in the derivation checks out.



One of the novel contributions of the paper is in using computationally cheap post-hoc local ensembles over fully trained ensembles in the baselines that require complicated training procedure. The other key differentiation over baselines is their method leverages the ill conditioned Hessian where  the baselines struggle in requiring an inverse of that ill conditioned Hessian.

The limitations of their method is in the determination of sufficiently small eigenvalues from the ensemble subspace. Further, the sensitivity of the small set of eigenvalues towards overestimating the prediction’s sensitivity to loss preserving perturbations and being less sensitive to some other under-constrained directions.
Below are the potential places where more clarity will help:
It would have been good to see a way of measuring the sensitivity in the set of small eigenvalues determination. I urge the authors to think of a way  of quantifying this sensitivity if possible especially since the model class is low dimensional.

In the experimental section with label, class prediction task, how correlated are the confounders Eyeglasses and Hat? What happens if the models are allowed to train for a longer; does the inconsistency in the behavior of AUC over more eigenvalues change?

In section 5.4, I think a comparison with the Resampling Under Uncertainty baselines is imperative.
There is a typo in E.4 label Attribute->Attractive.

There is lack of clarity in how similar the models are during training. Although, the ensemble is used post-hoc, its unclear if the models during training differ in initialization only?

What are the implications of the method in the case of finetuning models especially is the training data available for fine-tuning is low.

Further, is there any notion of how the method scales with increasing depth in the neural network models? A comparison with larger test set and models trained on deeper architecture such as ResNet and the like will be interesting to see.

With noisier data and the inconsistencies in the expected behavior of the method, is there a way of quantifying the amount of noise and the extrapolation? I do see the empirical experiments demonstrating this but some more insight into this is perhaps important.

Overall, its an extremely well written paper with great clarity. The method described by the authors is well differentiated from the baselines in making the clever use of the projection of the ill conditioned Hessian on the low curvature directions of the test point’s gradient.


**Experience Assessment:**

I have read many papers in this area.

**Review Assessment: Checking Correctness Of Derivations And Theory:**

I assessed the sensibility of the derivations and theory.

**Review Assessment: Checking Correctness Of Experiments:**

I assessed the sensibility of the experiments.

**Review Assessment: Thoroughness In Paper Reading:**

I read the paper thoroughly.

---

> ### Author Response · Authors · 2019-11-14
> **Response to Reviewer 2**
>
> We thank the reviewer for their thorough engagement with our paper. The reviewer raises several valid and interesting points that have helped us strengthen the paper and inspired us to consider extensions. We address specific points below.
>
> # Major Points
>
> > Measuring sensitivity to choice of eigenvalue cutoff
>
> We agree that this is an important point, and it was brought up by other reviewers as well. For our experiments in this paper, as a proof of concept, we fixed m to be 2000 for the tabular datasets, and m = 3000 for the CelebA dataset. In response to reviewer comments, we tried some other heuristics for setting cutoffs and found our results to be relatively insensitive.
>
> This being said, we would ideally aim for the largest subspace such that the curvature of the loss surface in any direction in that subspace is smaller than some tolerance. Because we use a method that iteratively calculates the highest-magnitude eigenvectors, we can (up to numerical error) upper bound the curvature in any single direction in the ensemble subspace by the magnitude of the m-th eigenvalue. This can be used to develop heuristics for what constitutes an ensemble subspace that is “flat enough.” For example, one could choose m such that a parameter perturbation of a given norm does not change the training loss by a given percentage. We have added some of this discussion to the body of the paper in Section 3. We may attempt to implement such a heuristic for the camera ready version of the paper for consistency.
>
> # Minor Points
>
> > “In the experimental section with label, class prediction task, how correlated are the confounders Eyeglasses and Hat? What happens if the models are allowed to train for a longer; does the inconsistency in the behavior of AUC over more eigenvalues change?”
> Eyeglasses and Hat have Pearson correlation coefficient 0.07 - they are not highly correlated.
>
> It is not clear to us if training models longer has a consistent effect on the results: we ran some experiments with the smaller models and observed inconclusive results. We agree that exploring these dynamics throughout training is definitely an interesting direction for future work.
>
> > “In section 5.4, I think a comparison with the Resampling Under Uncertainty baselines is imperative.”
>
> Unfortunately, Resampling Uncertainty Estimation (RUE) here is not well-defined for the active learning task: influence functions are intended to be calculated for model parameters which are trained to a local (convex) minimum. In active learning, this assumption does not hold since the parameters are not fully trained at each step. Additionally, implementation of RUE on larger models is non-trivial (note that no image models were evaluated in the RUE paper). This is because of the issues discussed in Section 4.1.
>
> > “There is lack of clarity in how similar the models are during training. Although, the ensemble is used post-hoc, its unclear if the models during training differ in initialization only?”
>
> Assuming you are referring to the ensemble experiments (e.g. Fig 2). Yes, the models are identical, differing in initialization only (specifically, model random seed changed. Data random seed was held constant).
>
> > “What are the implications of the method in the case of finetuning models especially is the training data available for fine-tuning is low.”
>
> This method could be quite useful in the case of finetuning, since this is a specific case where the model’s performance in the new (finetuning) domain could likely be underspecified due to a lack of relevant data in the original (non-finetuning) training set.
>
> > “Further, is there any notion of how the method scales with increasing depth in the neural network models? A comparison with larger test set and models trained on deeper architecture such as ResNet and the like will be interesting to see.”
>
> There is no reason why depth in particular would be an issue for our method. We agree that scaling up to models with larger parameter sizes is an important direction for improvement. However, in our current work we demonstrate that our approach can be successful on non-trivial models, and while we do not present results on SOTA image models, we believe our approach is still promising, and worthy of further investigation.
>
> > “With noisier data and the inconsistencies in the expected behavior of the method, is there a way of quantifying the amount of noise and the extrapolation? I do see the empirical experiments demonstrating this but some more insight into this is perhaps important.”
>
> I am a little unclear on the exact meaning of this question, but what I think you’re asking about is how noise in the data affects our method’s ability to detect extrapolation (as in Section 5.2). The place where noisy data might present the largest challenge would be in the minibatch calculation of the HVP. In this case, one can make a tradeoff with time if necessary - increasing the number of minibatches used to estimate the HVP.

---

### Official Review · AnonReviewer1 · 2019-11-03
**Official Blind Review #1**

**Rating:** 6

**Review:**

This paper presents local ensembles, a method for detecting underdetermination when extrapolating to test points. The authors define an extrapolation score which is used to estimate the standard deviation of predictions at test points. The extrapolation score is chosen to represent the variability in predictions that would be generated by models with similar training loss. By considering the eigenvectors of the Hessian that are associated with minimum eigenvalues the directions of the loss surface with minimal curvature are found, and perturbations of the parameters in the subspace of minimal curvature correspond to models with similar training loss.

The authors show that their extrapolation score is proportional to the first order approximation to the change in prediction under a perturbation of the parameters with minimal change in loss. In practice the minimum eigenvalue/eigenvector pairs are computationally challenging to compute for large matrices. For this reason the subspaces with minimal change in loss are computed by finding sets of vectors that are mutually orthogonal to the eigenvectors associated with dominant eigenvalues of the Hessian.

The method is validated experimentally first through out-of-distribution detection on synthetic data. The authors then test performance by constructing a "blind spot" by generating features that are a linear combination of existing features. Data can be generated as either within or out of distribution and the AUC metric can be applied to test model performance. In the final experiment the authors demonstrate the use of local ensembles in active learning. By determining which of the training samples are in the model's blind spots at each iteration and training based on these examples rather than randomly selecting training examples rates of convergence can be increased.

I vote to accept this paper as the proposed local ensemble method builds on a growing body of literature regarding loss-surface inference, providing a new way to connect the shape of the loss surface to extrapolation detection. The theoretical result showing the first order relationship between the standard deviation of extrapolation predictions and perturbations in solutions is a useful insight.

There are some points that should be addressed for clarity however. Firstly the proof of proposition 1 should be made clearer. This is central to the work of the paper and a more full treatment of the proof here could help illuminate some intuition about the connection to perturbations and variance of predictions.

The other main point that is not addressed is that in principal we aim to find the subspace associated with minimal eigenvalues, but in practice this is computationally prohibitive. Therefore a space that has a basis that is mutually orthogonal to the dominant eigenvectors is sought (the found subspace), and this could have minimal relation to the subspace that is actually sought (the optimal subspace). Some experimentation showing how the found subspace relates to the optimal subspace would be informative, as well as how sensitive the results are to how much the found and optimal subspaces differ.

Some minor points:
- Many of the plots lack axis labels, although many are explained in the captions the figure labeling needs to be improved
- Some explanation about the choice of AUC as a metric would be informative and could help connect to the initial motivation of the method
- Experiment details should be given in the main body of the paper rather than the appendix; i.e. in section 5.2 it is only explained that a "neural network" is trained, the architecture should be specifically given alongside the discussion of the experiment


**Experience Assessment:**

I have read many papers in this area.

**Review Assessment: Checking Correctness Of Derivations And Theory:**

I assessed the sensibility of the derivations and theory.

**Review Assessment: Checking Correctness Of Experiments:**

I assessed the sensibility of the experiments.

**Review Assessment: Thoroughness In Paper Reading:**

I read the paper at least twice and used my best judgement in assessing the paper.

---

> ### Author Response · Authors · 2019-11-14
> **Response to Reviewer 1**
>
> We thank the reviewer for their thorough engagement with our paper. The reviewer raises several interesting points that have helped us strengthen the paper. We address specific points below.
>
> # Major Points
>
> > Clarity of Proposition 1
>
> We have reworded the statement and proof of Proposition 1 to make the thinking a little more transparent. Please do let us know if there is still some ambiguity here.
>
> > Relation between found and optimal ensemble subspaces
>
> This is an important question. The notion of an “optimal” subspace could be formalized as the largest subspace such that the curvature of the loss surface in any direction in that subspace is smaller than some tolerance. Because we use a method that iteratively calculates the highest-magnitude eigenvectors, we can (up to numerical error) upper bound the curvature in any single direction in the ensemble subspace by the magnitude of the m-th eigenvalue. This can be used to develop heuristics for what constitutes an ensemble subspace that is “flat enough.” For example, one could choose m such that a parameter perturbation of a given norm does not change the training loss by a given percentage. We have added some of this discussion to the body of the paper in Section 3.
>
> For our experiments in this paper, as a proof of concept, we fixed m to be 2000, and found that in most cases, the corresponding eigenvalue was small (for the tabular experiments < 1e-4). A more thorough analysis would factor numerical error into this determination.
>
> # Minor Points
>
> > “Many of the plots lack axis labels, although many are explained in the captions the figure labeling needs to be improved”
>
> This is a good point and we have adjusted the figures in Sections 5.2 and 5.4 accordingly.
>
> > “Some explanation about the choice of AUC as a metric would be informative and could help connect to the initial motivation of the method”
>
> We chose to focus on this OOD detection task because this is that task that most previous methods have proposed. This being said, tasks with a more continuous notion of ground truth would also be useful to consider.
>
> We have added a sentence to Sec 5.1 commenting that we use AUC since OOD data may be rare.
>
> > “Experiment details should be given in the main body of the paper rather than the appendix; i.e. in section 5.2 it is only explained that a "neural network" is trained, the architecture should be specifically given alongside the discussion of the experiment”
>
> We agree with the reviewer that experimental details should go in the main body - however, we are somewhat space-constrained by the ICLR page limit. We have added a sentence to each experiment subsection stating the architecture of each model.

---

### Author Response · Authors · 2019-11-14
**(Brief) General Note - Typo Correction**

The value 0.738 for the MaxProb baseline, M/E task was erroneously bolded in Table 2, due to an error in confidence interval calculation. We have un-bolded it in the new draft. This means that the baseline is *not* in the 95% CI around the LE (Loss) performance (i.e. a significant improvement in performance by our method exists on this task).

---

### Decision · Program_Chairs · 2019-12-19

**Decision:**

Accept (Poster)

**Comment:**

This paper presents an ensembling approach to detect underdetermination for extrapolating to test points. The problem domain is interesting and the approach is simple and useful. While reviewers were positive about the work, they raised several points for improvement. The authors are strongly encouraged to include the discussion here in the final version.